# Effects of the COVID-19 pandemic on life expectancy and premature mortality in the German federal states in 2020 and 2021

Isabella Marinetti[1,2]*, Dmitri Jdanov[1,3], Pavel Grigoriev[4], Sebastian Klüsener[4,5,6], Fanny Janssen[2,7]

**1** Max Planck Institute for Demographic Research, Rostock, Germany, **2** Population Research Centre, Faculty of Spatial Sciences, University of Groningen, Groningen, The Netherlands, **3** National Research University Higher School of Economics, Moscow, Russia, **4** Federal Institute for Population Research (BiB), Wiesbaden, Germany, **5** University of Cologne, Cologne, Germany, **6** Vytautas Magnus University, Kaunas, Lithuania, **7** Netherlands Interdisciplinary Demographic Institute—KNAW/University of Groningen, The Hague, The Netherlands

* marinetti@demogr.mpg.de

**Data Availability Statement:** Data on raw death counts by federal state, age and sex, cannot be shared publicly because of restrictions of the data

## Abstract

The mortality impact of COVID-19 has mainly been studied at the national level. However, looking at the aggregate impact of the pandemic at the country level masks heterogeneity at the subnational level. Subnational assessments are essential for the formulation of public health policies. This is especially important for federal countries with decentralised health-care systems, such as Germany. Therefore, we assess geographical variation in the mortality impact of COVID-19 for the 16 German federal states in 2020 and 2021 and the sex differences therein. For this purpose, we adopted an ecological study design, using population-level mortality data by federal state, age, and sex, for 2005–2021 obtained from the German Federal Statistical Office. We quantified the impact of the pandemic using the excess mortality approach. We estimated period life expectancy losses (LE losses), excess premature mortality, and excess deaths by comparing their observed with their expected values. The expected mortality was based on projected age-specific mortality rates using the Lee-Carter methodology. Saxony was the most affected region in 2020 (LE loss 0.77 years, 95% CI 0.74;0.79) while Saarland was the least affected (-0.04, -0.09;0.003). In 2021, the regions with the highest losses were Thuringia (1.58, 1.54;1.62) and Saxony (1.57, 1.53;1.6) and the lowest in Schleswig-Holstein (0.13, 0.07;0.18). Furthermore, in 2021, eastern regions experienced higher LE losses (mean: 1.13, range: 0.85 years) than western territories (mean: 0.5, range: 0.72 years). The regional variation increased between 2020 and 2021, and was higher among males than among females, particularly in 2021. We observed an unequal distribution of the mortality impact of COVID-19 at the subnational level in Germany, particularly in 2021 among the male population. The observed differences between federal states might be partially explained by the heterogeneous spread of the virus in 2020 and by differences in the population's propensity to follow preventive guidelines.

owner, the German Federal Statistical Office. These data are available for everybody upon request for researchers who meet the criteria for access to confidential data and are part of a standard statistical report. The data underlying the results presented in the study are available from the German Federal Statistical Office, url: https://www.destatis.de/DE/Home/_inhalt.html. We included a link to the minimal dataset and scripts published on OSF (DOI 10.17605/OSF.IO/V2MRG).

**Funding:** The study was partially funded by the VolkswagenStiftung (https://www.volkswagenstiftung.de/en) grant "Strengthening a reliable evidence base for monitoring COVID-19 and other disasters". PG has received funding from the European Research Council (ERC), the European Union's Horizon 2020 (https://research-and-innovation.ec.europa.eu/funding/funding-opportunities/funding-programmes-and-open-calls/horizon-2020_en) research and innovation program (agreement No 851485). FJ has received funding from the Netherlands Organisation for Scientific Research (NWO, https://www.nwo.nl/en) (grant no. VIC.191.019). The funders had no role in study design, data collection and analysis, decision to publish, or preparation of the manuscript.

**Competing interests:** The authors have declared that no competing interests exist.

## Introduction

At the end of 2021, more than 5 million deaths attributable to COVID-19 have been estimated around the world [1].

Many studies have been conducted to understand the magnitude of the impact of the COVID-19 pandemic on mortality at the national level [1–7]. These national estimates, however, hide an uneven impact of the pandemic across subnational populations [8]. Indeed, previous studies have shown important inequalities in the mortality impact of COVID-19 by region [8–12], sex [13–15], socio-economic status [16–18], and race [17–19], and generally, higher case-fatality rates have been reported among older age groups [14, 20]. Still, however, more work is needed to obtain a detailed picture of the mortality impact of the COVID-19 pandemic. In the current study, we more closely examine regional variation, changes therein over time, and sex differences in the regional variation. We do so for Germany, using a validated analytical approach.

Studying the mortality impact of COVID-19 at the subnational level can provide essential public health information, particularly in countries where regions have a large degree of independence in healthcare decision-making. Moreover, examining the uneven spread of the COVID-19 pandemic can provide crucial information for strengthening the preparedness of healthcare systems for pandemics and health crises.

When measuring the health burden of the COVID-19 pandemic, several challenges arise, including determining the accuracy of cause-of-death statistics and capturing the direct and indirect effects of the pandemic on population health [21]. The excess mortality method, which is based on the comparison of observed and expected mortality, provides an objective and comparative way of assessing the impact of the pandemic [21, 22]. However, the accurate estimation of expected mortality is a key issue, as estimates can vary depending on the extrapolation method and the data used, and on the additional underlying choices and assumptions made [23–25]. Using suboptimal data or relying on questionable assumptions may lead to a significant under- or overestimation of the impact [25].

Germany provides an interesting case for studying the mortality impact of COVID-19 at the subnational level. First, Germany consists of federal states, which are mainly responsible for political decisions regarding public health, the regulation of healthcare interventions, hospitals, and the implementation of all healthcare measures [26]. Second, compared to other countries, Germany experienced relatively small losses in life expectancy, particularly in 2020 [4, 27], but also in 2021, even though the number of COVID-19 deaths was much higher in that year [28]. In 2020, life expectancy at birth in Germany was 0.38 years lower than expected for males and 0.20 years lower than expected for females. These figures correspond to approximately 18,000 excess male deaths and 12,000 excess female deaths [29].

Previous research on the impact of the COVID-19 pandemic on mortality in Germany was mostly restricted to the national level [30], albeit with some exceptions [31–33], and focused primarily on the first year of the pandemic. Substantial spatial variation in standardized mortality ratio excess was found during the first wave of the COVID-19 pandemic [34]. Most of these research also present methodological issues and limitations related to data availability. Zur Nieden and colleagues [35] used averages of weekly age-specific death counts for only the previous four years to estimate the expected number of deaths. Stang and colleagues [36] applied the same approach, but to total death rates, and using only the average of the previous four years in estimating the mortality baseline. Similarly, the results of advanced modelling performed by De Nicola [37] and König [38] and colleagues were based on data from 2016 to 2019. Although some of these mortality estimates guided public health policy responses to COVID-19 in Germany, the underlying methodology used led to an underestimation of the

mortality impact of COVID-19 [25]. The period considered for the benchmark mortality level, included two years of elevated mortality due to flu epidemics [39], leading to an over-estimation of expected mortality and, hence, an underestimation of the mortality impact of COVID-19.

In the current study, we analysed subnational variation in the mortality impact of COVID-19 for Germany in both 2020 and 2021, and the sex differences therein. Three different measures (losses in life expectancy at birth, excess premature mortality, excess deaths), were derived for this aim. They were constructed using a validated analytical approach [4] for the estimation of expected age- and sex-specific mortality. In doing so, we obtained more accurate and consistent estimates of the impact of the pandemic on mortality, not only for Germany as a whole but also for its 16 federal states. Overall, we found an unequal distribution of the mortality impact of COVID-19 at the subnational level in Germany, particularly among the male population, which increased over time.

## Data and methods

This study aims to assess the mortality impact of COVID-19 in Germany and its regions at the level of the 16 federal states (S1 Fig and SS1 Table) by sex in 2020 and 2021, using the excess mortality method [4]. This is the most robust and reliable approach to quantifying the mortality burden due to the pandemic [21]. The excess mortality was estimated as the difference between the expected and observed mortality.

For this purpose, we adopted an ecological study design, using population-level mortality data by federal state, age, and sex, for the years 2005–2021 obtained from the German Statistical Office. These data are available for everybody upon request and are part of a standard statistical report. Thus, this study doesn't require an ethical approval. The minimal dataset and the Rscript used for the computations are available on an OSF repository [40].

### Observed mortality in 2020 and 2021

Annual death counts and population exposures by federal state, sex, and single age for the years 2005–2020 were used (German Statistical Office) to estimate observed age-specific mortality rates. For 2021, annual death counts were not available and we estimated the observed age-specific death counts and population exposures by sex and federal state from the weekly deaths data published by Statistisches Bundesamt [41, 42]. However, the weekly death data were only available by broad age groups (0–64, 65–74, 75–79, 80+), hence we employed the methodology by Jdanov and colleagues [43] to split them into narrower age groups (0, 1–4, 5–9, . . ., 100+) using the age distribution of expected (forecasted) death counts.

### Expected mortality in 2020 and 2021

We estimated expected age-specific mortality rates by federal state and sex for 2020 and 2021 by forecasting the age-specific mortality rates by state and sex for the years 2005–2019 using the Lee-Carter mortality forecasting methodology [44]. Thus, in this way, we estimated the expected mortality levels for 2020 and 2021 in the absence of the pandemic [4]. We used the package demography [45] (v1.22) in R, and applied the forecasting model separately to each region and sex. The model was fitted by applying singular value decomposition (SVD), adjusting life expectancy at birth to correct for jump-off bias, and forecasting the $k_t$ term with a simple random walk with drift.

## Mortality indicators

Based on the observed and expected mortality measures for 2020 and 2021, we analysed three measures of excess mortality: losses in life expectancy at birth (LE losses), excess Years of Life Lost (excess YLL), and excess deaths. Life expectancy at birth is a widely used summary measure of mortality in a population, indicating the average number of years people can expect to live if subjected throughout the rest of their lives to the current mortality conditions.

We calculated LE losses as a difference between expected and observed life expectancy. Both observed and expected life expectancies were obtained by applying the Human Mortality Database (HMD) life table methodology [46] to, respectively, the observed and the expected (i.e., projected) age-specific mortality rates.

YLL is an often-used measure of premature mortality, which takes into account both the frequency of deaths and the ages at which deaths occur, giving greater weight to deaths at younger ages [47]. One YLL represents the loss of one year of life Both observed and expected YLL were calculated as:

$$YLL_{t,s,f} = \sum_{x} D_{x,t,s,f} * SLE_x$$

Hence, summing across age (*x*) the product of the number of observed or expected deaths (*D*) by age *x*, calendar year *t*, sex *s*, and federal state *f*, with a global standard life expectancy at the age at which deaths occur (*SLE_x*).

For this purpose, we used the WHO the global standard life expectancy by age [47]. In addition to the total excess YLL, we obtained the excess YLL per 100,000 population by dividing it by the (state-specific) population numbers.

Excess YLL was calculated as observed minus expected YLL. Excess deaths were computed as observed minus expected death counts.

## Confidence intervals

The confidence intervals around our estimates were computed using Monte Carlo simulations assuming that the mortality forecast was the only source of statistical uncertainty. The confidence intervals were based on a simulation of 5000 iterations developed in three steps: first, we derived the standard deviations of the age-specific expected mortality rates from the Lee-Carter model; second, we generated a random set of age-specific death rates; and third we computed the outcome measures of interest. The 2.5[th] and the 97.5[th] quantiles of the simulated distribution were used as 95% confidence intervals.

To assess the subnational variation and the sex differences therein, we computed the weighted standard deviation, the maximum and the minimum, and the range of the three measures analysed (LE losses, excess YLL, excess deaths).

## Results

### Regional patterns in life expectancy losses, years of life lost, and excess mortality

In 2020 in Germany, life expectancy decreased by 0.33 years (for 95% CI see S2 Table), and in terms of premature mortality, there were approximately 493,000 excess years of life lost (S3 Table). Moreover, the total number of excess deaths in Germany was around 600 deaths per 100,000 population (S4 Table). The largest effects of COVID-19 on mortality were observed in the eastern federal states of Saxony (LE loss: 0.77 years, excess YLL: 1,852 years per 100,000 population), Berlin (LE loss: 0.69 years, excess YLL: 1120 years per 100,000

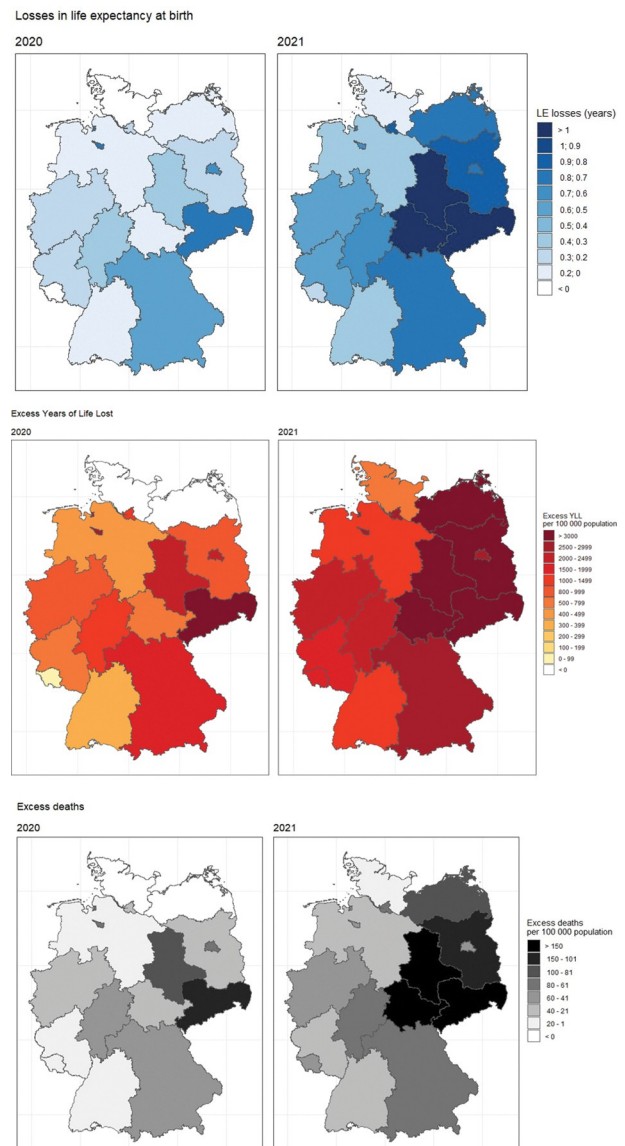

**Fig 1. Mortality impact of COVID-19 on German federal states in 2020 and 2021, both sexes combined.** (A) Losses in life expectancy at birth (LE losses), years. (B) Excess years of life lost (excess YLL) per 100,000 population. (C) Excess deaths per 100,000 population. Reprinted from GADM shapefile under a CC BY license, with permission from GADM, original copyright 2018–2022.

population), and Bavaria (LE loss: 0.4 years and excess YLL: 883 years per 100,000 population). The federal states where the impact of COVID-19 on mortality was smallest were Saarland (LE loss: -0.05 years, excess YLL: -846 years per 100,000 population) and Schleswig-Holstein (LE loss: -0.02 years, excess YLL: 96 years per 100,000 population) (Fig 1A and 1B).

In 2021 in Germany, life expectancy fell by 0.44 years, and there were approximately 686,000 excess years of life lost. The largest mortality impact of COVID-19 was observed in the eastern federal states of Thuringia (LE loss: 1.58 years, excess YLL: 3,943 years per 100,000 population) and Saxony (LE loss: 1.57 years, excess_YLL: 3,540 years per 100,000 population) (Fig 1A and 1B). Like in 2020, the federal states with the lowest losses were the western federal

**Table 1. Subnational variation in the mortality impact of COVID-19 and sex differences therein, across the 16 German federal states, 2020 and 2021.**

|  |  | Males | | Females | | Total | |
|---|---|---|---|---|---|---|---|
|  |  | **2020** | **2021** | **2020** | **2021** | **2020** | **2021** |
| **LE losses** | **Germany** | 0.41 | 0.56 | 0.23 | 0.26 | 0.33 | 0.44 |
|  | **Min** | 0.003 | 0.01 | -0.34 | 0.04 | -0.35 | 0.12 |
|  | **Max** | 0.87 | 1.83 | 0.64 | 1.23 | 0.79 | 1.57 |
|  | **Range** | 0.86 | 1.82 | 0.98 | 1.19 | 1.13 | 1.45 |
|  | **SD** | 0.27 | 0.51 | 0.28 | 0.36 | 0.31 | 0.44 |
| **Excess YLL (per 100,000)** | **Germany** | 800 | 1,825 | 318 | 804 | 1,118 | 2,629 |
|  | **Min** | -233 | 0.72 | -532 | 203 | -846 | 257 |
|  | **Max** | 2,333 | 5,050 | 1,494 | 2,874 | 1,852 | 3,840 |
|  | **Range** | 2,566 | 5,049 | 2,026 | 2,670 | 2,698 | 3,583 |
|  | **SD** | 705 | 1,483 | 529 | 674 | 827 | 1,000 |
| **Excess deaths (per 100,000)** | **Germany** | 44.6 | 83.3 | 27.8 | 50.5 | 72.3 | 133.8 |
|  | **Min** | -31.1 | 2.2 | -20.6 | 16.9 | -47.0 | 5.1 |
|  | **Max** | 155.6 | 312.5 | 139.6 | 198.4 | 139.9 | 217.5 |
|  | **Range** | 186.7 | 310.3 | 160.2 | 181.5 | 186.9 | 212.4 |
|  | **SD** | 50.1 | 87.5 | 39.6 | 56.0 | 44.3 | 58.5 |

LE losses: losses in life expectancy at birth; excess YLL: excess Years of Life Lost.

states of Schleswig-Holstein and Saarland (LE loss: 0.12 years, excess YLL: 626 years per 100,000 population; LE loss: 0.125, excess YLL: 257 years per 100,000 population).

Between 2020 and 2021 both for Germany as a whole and for every federal state, losses in life expectancy at birth and excess years of life lost increased. The increases varied however substantially across states. Losses in life expectancy at birth increased more in the eastern than in the western part of Germany. In particular, Mecklenburg-Western Pomerania and Thuringia completely lost their previous advantage. As such, in 2021, a much clearer East-West pattern in the mortality impact of COVID-19 emerged. Generally, the regional variation in the mortality impact of COVID-19 increased between 2020 and 2021 (Table 1).

Regarding excess deaths, the regional patterns in 2020 and 2021 displayed a similar pattern to the other indicators (Fig 1C and S4 Table).

## Sex differences in the regional patterns

Both the mortality impact of COVID-19 and its regional variation differed by sex (Table 1; S2–S4 Figs). Overall, males experienced higher excess mortality than females. Specifically, in Germany in 2020, males had almost double the values of LE losses (0.41 versus 0.23 years) and excess deaths (44.6 versus 27.8 per 100,000 population of females, and more than double the excess YLL of females (800 versus 391 per 100,000 population). These sex differences were slightly smaller in 2021 (LE losses: 0.53 and 0.33, respectively; excess deaths: 83 and 50 deaths per 100,000 population, respectively; excess YLL: 1,825 and 804 per 100,000 population, respectively).

In 2020, the regional variation in the mortality impact of COVID-19 did not differ greatly by sex. The regional variation in LE losses was approximately the same between males and females (SD: 0.28 and 0.27 years, respectively). We observed a larger regional variation in premature mortality among males than among females (SD excess YLL: 706 and 530 years, respectively), but the range was rather similar for males (2,567 years) and females (2,027 years). Furthermore, excess deaths varied slightly more between regions among males than among

females (SD: 50 and 39.6 deaths per 100,000 population, respectively; range: 186.7 and 160.2 deaths per 100,000 population, respectively).

In 2021, sex differences in the regional variation in excess mortality were substantially higher than in 2020. Compared to females, males had a 1.4 times higher SD in LE losses and a 0.6-year higher range. Similarly, the range of excess YLL among males was almost double that among females (5,049 compared to 2,671 years per 100,000 population). Moreover, we observed an increasing territorial variability of excess deaths between the female and male populations and the two years considered. Particularly in 2021, the regional variability of the excess death indicator was around 1.7 times higher among males than among females (range: 310.3 male deaths and 181.5 female deaths per 100,000 population, SD: 87.5 male deaths and 56 female deaths per 100,000 population).

## Discussion

### Main findings

This study analysed the mortality impact of the COVID-19 pandemic for Germany at the subnational level in 2020 and 2021 using three outcome measures (LE losses, excess YLL, excess deaths) and methodology that allowed us to compute consistent and more accurate estimates than before.

Overall, we observed an unequal distribution of the mortality impact of COVID-19 at the subnational level in Germany. Geographical differences in the mortality impact of COVID-19 were also observed before in other countries, such as Italy, Spain, England, Greece, and Switzerland [8, 11, 33, 48]. Because of different methodologies to compute expected death or different periods analysed, one-on-one comparisons of our observed geographical variation with those in these other countries cannot be made directly. However, our findings are in line with previous research [23, 24] and address robustness, bias and baseline approach issues. This notwithstanding, we can conclude that the variations that we observed among the German federal states are rather high, also considering that Germany was one of the countries with the lowest value of excess mortality at the beginning of the pandemic [49]. Another interesting finding is the increase in the regional variation between 2020 and 2021. However, due to the lack of literature on the analysis of COVID-19 spatial mortality over multiple years, it is still very difficult to make a comparison with findings for other countries. Indeed, in Germany, we observed an increase in the regional variation between 2020 and 2021, particularly among the male population. Saxony, Berlin, and Bavaria were the most affected federal states during the first year of the pandemic. In 2021, the mortality impact of COVID-19 was larger in almost all eastern federal states (Thuringia, Saxony, Saxony-Anhalt, and Brandenburg) than in other parts of Germany. Thus, a clear East-West pattern of the mortality impact of COVID-19 could be observed in 2021.

To explain the observed regional pattern, we considered four groups of risk factors: the spatial distribution of the virus in the country, the conditions of the state-specific healthcare systems, the set of measures introduced at the subnational level, and individual behavior regarding vaccination against COVID-19. Although we could not distinguish the influence of each individual factor, we found that the spatial variation was higher during periods when the pandemic burden was greater.

In 2020, the disparities in the mortality impact of COVID-19 within Germany could be explained by the heterogeneous spread of the virus among the federal states. At the subnational level, the number of deaths was significantly higher in the federal states that were first affected by the COVID-19 pandemic, like Bavaria [50]. On the other hand, in Saxony, the majority of deaths occurred in late 2020 and early 2021. These territories were mainly affected during the

winter wave of 2020 when the virus had already reached all the German regions. Indeed, it is likely that the patterns observed in the southeastern regions were due to their proximity to countries such as Poland and the Czech Republic, which experienced a sudden peak in mortality attributable to the COVID-19 pandemic during the autumn of 2020.

In 2021, we observed a different regional pattern, with greater regional variability. In particular, we found that the impact of COVID-19 on mortality was larger in most eastern German federal states. Compared to their western German counterparts, eastern Germans were more likely to live in more deprived districts and were more reluctant to adhere to containment measures. It thus appears that socioeconomic disparities in mortality related to COVID-19 emerged over the course of the second pandemic wave in Germany [51]. This can likely be explained by the fact that the incidence rate of COVID-19 infections was much higher in eastern Germany, potentially due to the lower vaccination rates and lower acceptance of containment measures in the East. Moreover, some differences in the mortality levels and the healthcare systems of eastern and western Germany still remain [52].

We also observed important sex differentials in the mortality impact of COVID-19 at both the national and the subnational levels. It appears that the male population was more affected by the pandemic and suffered the largest life expectancy losses in both years of the pandemic. This result confirms early findings for the first stage of the pandemic in several countries around the world [13, 14]. The sex differences in COVID-19 mortality may be explained not only by factors such as sex differences in immune responses, biological characteristics, and underlying comorbidities [13, 15] but also by sex differences in levels of exposure to the virus and willingness to follow health guidelines regarding vaccination or prevention.

In addition, we observed sex differences in the regional pattern of the mortality impact of COVID-19. Specifically, we found that the territorial variation was higher among males than among females in 2021, but not in 2020. This finding could be attributable to the overall increase in COVID-19 mortality in the second year of the pandemic, which may have exacerbated the regional inequalities between the male and female populations. Moreover, the observed sex differences in the mortality impact of COVID-19 were mainly in specific eastern federal states such as Brandenburg and Berlin, especially in 2021 (S2–S4 Figs), indicating that the sex differences in the mortality impact of COVID-19 are importantly influenced by the particular context. In recent decades, eastern German females have experienced larger health improvements than eastern German males at working ages [53], and it is likely that COVID-19-specific mechanisms have further increased the sex discrepancies in eastern Germany.

## Limitations of the study

Some limitations of the study must be considered. First, although we used standard and commonly used mortality measures, they have their own limitations. Period life expectancy at birth (LE), as a summary measure, refers to a synthetic cohort and does not reflect a real mortality experience of any cohort; the premature mortality measure, years of life lost (YLL), depends entirely on the population structure and the death counts; and the WHO standardised life expectancy (SLE) used for the calculation of YLL refers to a fictitious population and is used in the analysis merely to allow the comparison of our results to the existing literature. Second, when applying the excess mortality method to small numbers the obtained results are less robust and should be interpreted with caution. In our study, this applies particularly to the federal state of Bremen, with a population size of around 600 thousand inhabitants. Third, the method and fitting period used for the computation of the expected level of mortality–to compare the observed mortality level to it- should be carefully chosen to avoid an under- or overestimation of the estimates of the mortality impact of COVID-19. In our study, we avoided bias

as much as possible by using a long series (15 years) of historical data and by employing the benchmark Lee-Carter mortality projection method. Nevertheless, using a different projection method and a different fitting period may lead to slightly different results. Lastly, when COVID-19 deaths are not many it is harder to statistically isolate the mortality impact of the pandemic, independent of the method. Still, however, the second and last limitations are both (partly) inherent to subnational level studies and should not prevent from undertaking these analyses because of the additional insights they still bring.

## Conclusion

All in all, our study extends the existing literature on the effects of the COVID-19 pandemic at the subnational level. We found substantial regional variation in the mortality impact of the COVID-19 pandemic in Germany, particularly in 2021 in the male population, with a clear East-West pattern of the mortality impact of COVID-19 in 2021. The differences between the federal states are likely related to the heterogeneous spread of the virus in 2020, and the differences in the propensity of the population to follow preventive guidelines and get vaccinated in 2021. The regional differences in the mortality impact of COVID-19 we observed provide new insights into the effects of the pandemic on mortality patterns in Germany.

We hope that in future assessments of excess mortality, more attention will be devoted to the advantages and disadvantages of specific approaches to derive estimates of expected mortality. The observation that the overall East-West mortality gradient increased during the COVID-19 pandemic should be taken into consideration when developing plans for health-care responses for future pandemics, not only at the level of specific federal states but also at the national level. Further research should be carried out not only to understand the potential long-term impact of the COVID-19 pandemic on long-term longevity trends, but also on their effects on widening health and mortality inequalities within countries.

## Supporting information

**S1 Fig. Map of Germany and federal states, East-West division.** Reprinted from GADM shapefile under a CC BY license, with permission from GADM, original copyright 2018–2022.
(TIF)

**S2 Fig. Losses in life expectancy at birth by sex and by federal state in 2020 and 2021.** Reprinted from GADM shapefile under a CC BY license, with permission from GADM, original copyright 2018–2022.
(TIF)

**S3 Fig. Excess Years of Life Lost (excess YLL) per 100,000 population by sex and by federal state in 2020 and 2021.** Reprinted from GADM shapefile under a CC BY license, with permission from GADM, original copyright 2018–2022.
(TIF)

**S4 Fig. Excess deaths and excess deaths per 100,000 population, by sex and by federal state in 2020 and 2021.** Reprinted from GADM shapefile under a CC BY license, with permission from GADM, original copyright 2018–2022.
(TIF)

**S1 Table. Population exposures of German federal states, Germany, and percentage (%) of population.**
(DOCX)

**S2 Table. Table with losses in life expectancy at birth by sex and by federal state in 2020 and 2021.**
(DOCX)

**S3 Table. Table with excess YLL and excess YLL per 100,000 population, by federal state and by sex, 2020 and 2021.**
(DOCX)

**S4 Table. Table with excess deaths and excess deaths per 100,000 population, by federal state and by sex, 2020 and 2021.**
(DOCX)

## Acknowledgments

IM gratefully acknowledges the resources provided by the International Max Planck Research School for Population, Health and Data Science (IMPRS-PHDS).

## Author Contributions

**Conceptualization:** Isabella Marinetti, Dmitri Jdanov, Fanny Janssen.

**Data curation:** Isabella Marinetti, Pavel Grigoriev, Sebastian Klüsener.

**Formal analysis:** Isabella Marinetti.

**Funding acquisition:** Dmitri Jdanov.

**Investigation:** Isabella Marinetti.

**Methodology:** Isabella Marinetti, Dmitri Jdanov.

**Software:** Isabella Marinetti.

**Supervision:** Dmitri Jdanov, Fanny Janssen.

**Visualization:** Isabella Marinetti.

**Writing – original draft:** Isabella Marinetti.

**Writing – review & editing:** Isabella Marinetti, Dmitri Jdanov, Pavel Grigoriev, Sebastian Klüsener, Fanny Janssen.

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
