## [Decision Letter · Decision Letter 0]

3 Sep 2023

PONE-D-23-10425Effects of the COVID-19 pandemic on life expectancy and premature mortality in the German federal states in 2020 and 2021PLOS ONE

Dear Dr. Marinetti,

Thank you for submitting your manuscript to PLOS ONE. After careful consideration, we feel that it has merit but does not fully meet PLOS ONE’s publication criteria as it currently stands. Therefore, we invite you to submit a revised version of the manuscript that addresses the points raised during the review process. The manuscript has been evaluated by two reviewers, and their comments are available below. The reviewers have raised a number of major concerns. They request improvements to the reporting of methodological aspects of the study, as well as revisions to improve the discussion of the work in the context of the literature. Could you please carefully revise the manuscript to address all comments raised?

We look forward to receiving your revised manuscript.

Kind regards,

Marianne Clemence

Staff Editor

PLOS ONE

Journal Requirements:

4. We note that Figures 1A, 1B, 1C, S1, S2, S3 and S4 in your submission contain map/satellite images which may be copyrighted. All PLOS content is published under the Creative Commons Attribution License (CC BY 4.0), which means that the manuscript, images, and Supporting Information files will be freely available online, and any third party is permitted to access, download, copy, distribute, and use these materials in any way, even commercially, with proper attribution. For these reasons, we cannot publish previously copyrighted maps or satellite images created using proprietary data, such as Google software (Google Maps, Street View, and Earth). For more information, see our copyright guidelines: http://journals.plos.org/plosone/s/licenses-and-copyright.

a. You may seek permission from the original copyright holder of Figures 1A, 1B, 1C, S1, S2, S3 and S4 to publish the content specifically under the CC BY 4.0 license.  

Reviewers' comments:

Reviewer's Responses to Questions

**Comments to the Author**

1. Is the manuscript technically sound, and do the data support the conclusions?

Reviewer #1: Yes

Reviewer #2: Partly

2. Has the statistical analysis been performed appropriately and rigorously? 

Reviewer #1: Yes

Reviewer #2: Yes

3. Have the authors made all data underlying the findings in their manuscript fully available?

Reviewer #1: Yes

Reviewer #2: Yes

4. Is the manuscript presented in an intelligible fashion and written in standard English?

Reviewer #1: Yes

Reviewer #2: Yes

5. Review Comments to the Author

Reviewer #1: Thank you for conducting such a study, there are several things needed to be further clarified

1. In Data and Method section, state clearly the study design, the inclusion and exclusion criteria, the source of data, the time frame of the study, and the ethical permission to conduct this study

2. In the discussion section: write about the limitations of the study.

Reviewer #2: Researchers must follow the instructions of the journal

Introduction

The impacts of COVID_19 have also been captured in numerous studies that are appearing elsewhere, or that are currently under review. Please consider revising your manuscript to speak directly to the global literature - and debates about covid-19 and mortality. That would help you to better situate the original contribution of this research.

The introduction should briefly place the study in a broad context and highlight why it is important. It should define the purpose of the work and its significance. The authors should review the current state of the research and document key publications. Please highlight controversial and diverging hypotheses when necessary. Finally, briefly mention the main aim of the work and highlight the principal conclusions.

Methodology

The methodology section is disorganized and not carefully written, the Materials and Methods should be described with sufficient details to allow others to replicate and build on the published results.

Results

The symbols are written incorrectly as the life expectancy. e0

discussion

Authors should discuss the results and how they can be interpreted from the perspective of previous studies and of the working hypotheses. The findings and their implications should be discussed in the broadest context possible. Future research directions may also be highlighted.

6. PLOS authors have the option to publish the peer review history of their article (what does this mean?). If published, this will include your full peer review and any attached files.

Reviewer #1: No

Reviewer #2: No

---

## [Author Response · Author response to Decision Letter 0]

14 Oct 2023

Response to Reviewers 

Dear Academic Editor, thank you for giving us the opportunity to submit a revised draft of the manuscript “Effects of the COVID-19 pandemic on life expectancy and premature mortality in the German federal states in 2020 and 2021” for publication in PLOS ONE. We appreciate the time and effort that you and the reviewers dedicated to providing feedback on our manuscript.

We have incorporated most of the suggestions made by the reviewers, throughout the whole manuscript. Those changes are highlighted within the manuscript. Please see below the questions/comments raised by the reviewers (in italics) and our responses. All page numbers refer to the revised manuscript file with tracked changes.

Reviewer #1:

Thank you for conducting such a study, there are several things needed to be further clarified

Author response: Thank you. Please see the below revision to the manuscript according to your comments.

1. In Data and Method section, state clearly the study design, the inclusion and exclusion criteria, the source of data, the time frame of the study, and the ethical permission to conduct this study

Authors response: This is an ecological study. Thus, we are using population-level mortality data provided by the German statistical office. There is no need for ethical permission. The Data and Method section has been updated to clarify this, and we also added a sentence on our study design and the data source in our abstract. Thank you for the advice.

Pages 5-6: This study aimed to assess the mortality impact of COVID-19 in Germany and its regions at the level of the 16 federal states (S1 Fig and S1 Table) by sex in 2020 and 2021, using the excess mortality method [4]. This is the most robust and reliable approach to quantifying the mortality burden due to the pandemic [21]. The excess mortality was estimated as the difference between the expected and observed mortality. For this purpose, we adopted an ecological study design, using population-level mortality data by federal state, age, and sex, for the years 2005-2021 obtained from the German Statistical Office. These data are available for everybody upon request and are part of a standard statistical report. No ethical approval is required. 

 2. In the discussion section: write about the limitations of the study.

Authors response: Thank you for the suggestion. We included a limitation section in the discussion. 

Page 16: Limitations of the study

Some limitations of the study must be considered. First of all, when applying the excess mortality method to small numbers the obtained results are less robust and should be interpreted with caution. In our study, this applies particularly to the federal state of Bremen, with a population size of around 600 thousand inhabitants. Second, the method and fitting period used for the computation of the expected level of mortality – to compare the observed mortality level to it- should be carefully chosen to avoid an under- or overestimation of the estimates of the mortality impact of COVID-19. In our study, we avoided bias as much as possible by using a long series (15 years) of historical data and by employing the benchmark Lee-Carter mortality projection method. Nevertheless, using a different projection method and a different fitting period may lead to slightly different results. Third, when COVID-19 deaths are not many it is harder to statistically isolate the mortality impact of the pandemic, independent of the method. Still, however, the first and third limitations are both (partly) inherent to subnational level studies and should not prevent from undertaking these results because of the additional insights they still bring.

Reviewer #2:

Researchers must follow the instructions of the journal

Authors response: The manuscript has been revised to meet the PLOS ONE style requirements.

Introduction

The impacts of COVID_19 have also been captured in numerous studies that are appearing elsewhere, or that are currently under review. Please consider revising your manuscript to speak directly to the global literature - and debates about covid-19 and mortality. That would help you to better situate the original contribution of this research.

The introduction should briefly place the study in a broad context and highlight why it is important. It should define the purpose of the work and its significance. The authors should review the current state of the research and document key publications. Please highlight controversial and diverging hypotheses when necessary. Finally, briefly mention the main aim of the work and highlight the principal conclusions.

Authors response: Thank you for the comment. The paper is focused on spatial inequality in COVID-19 burden at subnational level. Due to limited data availability, there is lack of publications on this topic. Nevertheless, we agree that providing more info on the global context helps to better situate the manuscript’s contribution to the existing COVID-19 literature. Thus, we revised the introduction in response to the comments. In particular, we added several key publications on the broad topic of COVID-19 and mortality to the Introduction section.

Page 3: “At the end of 2021, more than 5 million deaths attributable to COVID-19 have been estimated around the world [1]. 

Many studies have been conducted to understand the magnitude of the impact of the COVID-19 pandemic on mortality at the national level [1-7]. These national estimates, however, hide an uneven impact of the pandemic across subnational populations [8]. Indeed, previous studies have shown important inequalities in the mortality impact of COVID-19 by region [8-12], sex [13-15], socio-economic status [16-18], and race [17-19], and generally, higher case-fatality rates have been reported among older age groups [14, 20]. Still, however, more work is needed to obtain a detailed picture of the mortality impact of the COVID-19 epidemic.”

Moreover, in the final paragraph of the introduction [page 5], we have clarified the objective of the study and included the overall findings:

“Overall, we found an unequal distribution of the mortality impact of COVID-19 at the subnational level in Germany, particularly among the male population, which increased over time.”

Methodology

The methodology section is disorganized and not carefully written, the Materials and Methods should be described with sufficient details to allow others to replicate and build on the published results

Authors response: We have revised the data and methods section to allow a better understanding of the underlying data and the methodologies used to obtain the presented results [pages 5-7]. In particular, we provide a brief overview of the methodology and data:

Page 5-6: This study aims to assess the mortality impact of COVID-19 in Germany and its regions at the level of the 16 federal states (S1 Fig and S1 Table) by sex in 2020 and 2021, using the excess mortality method [4]. This is the most robust and reliable approach to quantifying the mortality burden due to the pandemic [21]. The excess mortality was estimated as the difference between the expected and observed mortality. For this purpose, we adopted an ecological study design, using population-level mortality data by federal state, age, and sex, for the years 2005-2021 obtained from the German Statistical Office. These data are available for everybody upon request and are part of a standard statistical report. No ethical approval is required.”

We restructured the provided information, and – where necessary - added details to allow others to replicate and build on the published results. In particular, we have divided the Data & Methods section into subsections (“Observed mortality in 2020 and 2021”, “Expected mortality in 2020 and 2021”, “Mortality indicators”, and “Confidence intervals”), reorganized all 4 subsections to make the results replicable and added the YLL formula.

Moreover, the minimal dataset and the Rscript used for the computations are available on an OSF repository (DOI 10.17605/OSF.IO/V2MRG).

Results

The symbols are written incorrectly as the life expectancy. e0

Authors response: Thank you for pointing this out. The life expectancy symbol has now been changed into LE (and e0_loss into LE_ loss) throughout the whole manuscript, as also used in other similar papers (see Schöley et al. 2022 https://www.nature.com/articles/s41562-022-01450-3). Moreover, the previous symbol for excess Years of Life Lost has been changed from Excess_YLL to Excess YLL for better readability of the manuscript.

discussion

Authors should discuss the results and how they can be interpreted from the perspective of previous studies and of the working hypotheses. The findings and their implications should be discussed in the broadest context possible. Future research directions may also be highlighted.

Authors response: As suggested by the reviewer, we have reviewed the discussion part, dividing it into: “Main findings”, “Limitations of the study”, and “Conclusions”. Furthermore, we have added a comparison of our results with previous studies on the mortality impact of COVID-19 at the subnational level (paragraph 2 of the Discussion). We now placed our results in the wider context of the effect of COVID on growing sub-national inequalities and lastly, we added future research directions in line with this:

Page 17: “Further research should be carried out not only to understand the potential long-term impact of the COVID-19 pandemic on long-term longevity trends, but also on their effects on widening health and mortality inequalities within countries.”

---

## [Decision Letter · Decision Letter 1]

3 Nov 2023

PONE-D-23-10425R1Effects of the COVID-19 pandemic on life expectancy and premature mortality in the German federal states in 2020 and 2021PLOS ONE

Dear Dr. Marinetti,

Thank you for submitting your manuscript to PLOS ONE. After careful consideration, we feel that it has merit but does not fully meet PLOS ONE’s publication criteria as it currently stands. Therefore, we invite you to submit a revised version of the manuscript that addresses the points raised during the review process.

We look forward to receiving your revised manuscript.

Kind regards,

Claudio Alberto Dávila-Cervantes, Ph.D.

Academic Editor

PLOS ONE

Journal Requirements:

Reviewers' comments:

Reviewer's Responses to Questions

**Comments to the Author**

1. If the authors have adequately addressed your comments raised in a previous round of review and you feel that this manuscript is now acceptable for publication, you may indicate that here to bypass the “Comments to the Author” section, enter your conflict of interest statement in the “Confidential to Editor” section, and submit your "Accept" recommendation.

Reviewer #1: All comments have been addressed

Reviewer #3: (No Response)

2. Is the manuscript technically sound, and do the data support the conclusions?

Reviewer #1: Yes

Reviewer #3: Yes

3. Has the statistical analysis been performed appropriately and rigorously? 

Reviewer #1: Yes

Reviewer #3: Yes

4. Have the authors made all data underlying the findings in their manuscript fully available?

Reviewer #1: Yes

Reviewer #3: No

5. Is the manuscript presented in an intelligible fashion and written in standard English?

Reviewer #1: Yes

Reviewer #3: Yes

6. Review Comments to the Author

Reviewer #1: All required data and related information are included in the manuscript.

Thank you to correcting the manuscript as required.

Reviewer #3: I have very minor comments to the authors since this seems to be a revised version already. Some sentenced read oddly and would argue there are two additional limitations:

They could also describe limitations of the indicators: LE referring to a synthetic cohort, YLL being affected by population structure and death counts, SLE may pertain to a fictitious population and not Germany, and while they argue that their methodology is more accurate than others they do not present evidence of this. For example, how do results compare with methods used by Shoeley or Nepomuceno et al. ?

here are some minor comments:

Abstract: ‘Therefore, assess geographical… therein’ This sentence does not make sense, is lacking a purpose.

Replace COVID-19 epidemic with COVID-19 pandemic

The sentence ‘Most of these research experienced methodological issues..’ reads oddly, perhaps rephrasing to ‘Mos to these research presents methodological limitations…

The phrase ‘advanced methodology’ is inaccurate. I would argue that the Lee-Carter method is standard and well known method.

7. PLOS authors have the option to publish the peer review history of their article (what does this mean?). If published, this will include your full peer review and any attached files.

Reviewer #1: No

Reviewer #3: No

---

## [Author Response · Author response to Decision Letter 1]

27 Nov 2023

Dear Dr. Dávila-Cervantes, 

Thank you for giving us the opportunity to submit a revised draft of the manuscript “Effects of the COVID-19 pandemic on life expectancy and premature mortality in the German federal states in 2020 and 2021”, for reconsideration for publication PLOS ONE. We would like to thank you and the reviewers for suggesting minor, although constructive, revisions to our manuscript. The recommendations and advice have helped us enhance the manuscript's quality. In particular, in accordance with your request, the reference list has been modified to ensure that it is complete and correct, visible in tracked changes of the manuscript.

In accordance with Reviewer 3#’s comments, we have modified the conclusions and limitations of the study adding two more points. Moreover, we have added the reference to the data and scripts OSF repository in the Data and Methods section (page 5). Our revisions to the text are highlighted within the manuscript. Our point-by-point responses to the reviewers’ comments are shown below after the questions/comments raised by the reviewers (in italics). All page numbers refer to the revised manuscript file with tracked changes.

Reviewer #1: 

1. All required data and related information are included in the manuscript.

Thank you to correcting the manuscript as required.

Author response: Thank you for your suggestions and the time spent on the revision of the manuscript.

Reviewer #3: 

1.I have very minor comments to the authors since this seems to be a revised version already. Some sentenced read oddly and would argue there are two additional limitations:

They could also describe limitations of the indicators: LE referring to a synthetic cohort, YLL being affected by population structure and death counts, SLE may pertain to a fictitious population and not Germany, and while they argue that their methodology is more accurate than others they do not present evidence of this. For example, how do results compare with methods used by Shoeley or Nepomuceno et al. ?

Author response: Thank you for your comments. The manuscript has been revised according to your suggestions. Particularly:

Page 14: “First, although we used standard and commonly used mortality measures, they have their own limitations. Period life expectancy at birth (LE), as a summary measure, refers to a synthetic cohort and does not reflect a real mortality experience of any cohort; the premature mortality measure, years of life lost (YLL), depends entirely on the population structure and the death counts; and the WHO standardised life expectancy (SLE) used for the calculation of YLL refers to a fictitious population and is used in the analysis merely to allow the comparison of our results to the existing literature.” 

Secondly, the sentence “However, our findings are in line with previous research, using different methods [Scholey 2021, Nepomuceno et al. 2022] and address robustness, bias and baseline approach issues.” [page 12] has been added to frame our findings and methodology in the context of validation and justification of the methods used in the analyses.

2.here are some minor comments: Abstract: ‘Therefore, assess geographical… therein’ This sentence does not make sense, is lacking a purpose. Replace COVID-19 epidemic with COVID-19 pandemic. The sentence ‘Most of these research experienced methodological issues..’ reads oddly, perhaps rephrasing to ‘Mos to these research presents methodological limitations…

The phrase ‘advanced methodology’ is inaccurate. I would argue that the Lee-Carter method is standard and well known method.

Author response: We have modified the manuscript according to your suggestions.

---

## [Editor Report · Decision Letter 2]

29 Nov 2023

Effects of the COVID-19 pandemic on life expectancy and premature mortality in the German federal states in 2020 and 2021

PONE-D-23-10425R2

Dear Dr. Marinetti,

We’re pleased to inform you that your manuscript has been judged scientifically suitable for publication and will be formally accepted for publication once it meets all outstanding technical requirements.

Kind regards,

Claudio Alberto Dávila-Cervantes, Ph.D.

Academic Editor

PLOS ONE

---

## [Editor Report · Acceptance letter]

13 Dec 2023

PONE-D-23-10425R2 

PLOS ONE

Dear Dr. Marinetti, 

I'm pleased to inform you that your manuscript has been deemed suitable for publication in PLOS ONE. Congratulations! Your manuscript is now being handed over to our production team.

Kind regards, 

on behalf of

Mr. Claudio Alberto Dávila-Cervantes 

Academic Editor

PLOS ONE